# Transplant Drugs against SARS, MERS and COVID-19

**René Hage [1,2,*]**, **Carolin Steinack [1,2]**, **Fiorenza Gautschi [1,2]** and **Macé M. Schuurmans [1,2]**

[1]   Division of Pulmonology, University Hospital Zurich, Raemistrasse 100, 8091 Zurich, Switzerland;
     carolin.steinack@usz.ch (C.S.); fiorenza.gautschi@usz.ch (F.G.); mace.schuurmans@usz.ch (M.M.S.)
[2]   Faculty of Medicine, University of Zurich, Raemistrasse 71, 8006 Zurich, Switzerland
*   Correspondence: rene.hage@usz.ch

**Abstract:** There is an urgent need to develop drugs and vaccines to counteract the effects of the new coronavirus SARS-CoV-2 and adequately treat the corona virus disease (COVID-19). As these drugs are still under investigation, research also focuses on existing medication with proven effectiveness in other coronaviral diseases. The advantages of existing therapeutic drugs that are currently approved (for other indications) are the known safety profile, general availability and relatively lower costs involved in extending the purpose to a new disease. Calcineurin inhibitors (CNI) are drugs that have shown effectiveness in several coronaviral diseases, and are well-known and widely used drugs in transplant medicine. The aim of this narrative review is to present the current evidence of CNI in coronaviral diseases, the biophysiology of CNI and to suggest possible ways to study CNI as a new treatment option for COVID-19. We searched original papers, observational studies, case reports, and meta-analyses published between 2000 and 2020 in English in the PubMed database and Google Scholar using the keywords: (coronavirus), (treatment), (MERS), (SARS), (COVID-19), (tacrolimus), (ciclosporin), (cyclosporin) AND (calcineurin inhibitor). We excluded studies in patients with clear indications for immunosuppressive therapy. Additionally, we searched in the preprint servers and the World Health Organization bulletin. Ten studies were identified and included. Calcineurin inhibitor therapy has been suggested to be effective for coronaviral diseases in different settings. The results are summarized in a table. CNI should be investigated as a first treatment option based on evidence of direct antiviral effects and its properties preventing severe systemic hyperinflammation, as has been observed in COVID-19 with predominantly pulmonary immunopathological changes.

**Keywords:** immunosuppression; treatment; hypothesis; cytokine storm syndrome; hyperinflammation; tacrolimus; FK506; cyclosporine

## 1. Introduction

Coronaviruses (CoV) are among the frequent pathogens causing the common cold. They have a single-stranded RNA genome, that is coiled within the virion. In electron microscopy they show spikes protruding from the virion envelope with a crown-like shape, which lead to the name "coronavirus".

They belong to the order of the Nidovirales, and within this order, the coronaviruses have been studied in great detail because of their zoonotic transmission since the 21st century, causing life-threatening infections in humans, their societal and economic impact, unusual features of their pathogenesis, and the complexity of their molecular biology [1]. The coronaviruses are classified into two main subfamilies: the Torovirinae and the Coronavirinae, the latter being subdivided into the genera Alpha-, Beta-, Gamma-, and Deltacoronavirus [1]. The Alpha- and Betacoronaviruses include the seven Coronavirus serotypes, of which there are four (CoV-NL63, -HKU1, -E229, -OC43) with a low pathogenicity, causing mild upper respiratory tract infections. The other three serotypes are highly dangerous viruses, such as the Severe Acute Respiratory Syndrome

Coronavirus-1 (SARS-CoV-1) causing SARS, the Middle East Respiratory Syndrome coronavirus (MERS-CoV) causing MERS, and the novel SARS-CoV-2 causing Coronavirus Disease-19 (COVID-19). So far, Gamma- and Deltacoronaviruses have been discovered mostly in avian species [1]. The Gamma- and Deltacoronaviruses cause economically important diseases of livestock, poultry, and laboratory rodents [2].

## 1.1. Pathophysiology in Coronaviral Infections

All coronaviruses have a different antigenicity, depending on the spike (S-) protein of the virus. In contrast to influenza viruses, the S-proteins in coronaviruses are very stable. To enter the human cell, coronaviruses use different human cell surface peptidases. Both SARS-CoV and SARS-CoV-2 use the human angiotensin-converting enzyme-2 (ACE-2), which functions as a receptor for the virus. This receptor is widely expressed in a number of organs including pulmonary tissue, as well as in monocytes and macrophages [3]. Recent studies also demonstrated that both SARS-CoV and SARS-CoV-2 also can use lectins to enter the cell. Moreover, SARS-CoV-2 can use neuropilin-1, which is strongly expressed by endothelial cells and epithelial cells facing the nasal cavity [4–6].

Entering the cytoplasm by the receptor, the virus uncoats and starts replicating in the human cell. The exact pathophysiology responsible for the unusually high morbidity and mortality following CoV infections with high pathogenicity, are incompletely understood. Important mechanisms could be the virus-induced direct cytopathic effects, as well as the viral evasion of the host immune system.

## 1.2. Morbidity and Mortality of Coronaviruses

A dysregulated immune system, resulting in an overshooting inflammatory response, contributes to morbidity and mortality. Mortality rates in MERS, SARS and COVID-19 are around 35%, 9% and 5% of infected individuals, respectively. Nevertheless, the number of infected patients has never been so large as in the current COVID-19 pandemic. The total number of patients suffering from MERS was 2400, from SARS 8300, and from COVID-19 (so far) passes 17 million [7]. MERS spread to 27 countries, SARS to 30 countries and COVID-19 represents currently a global threat of increasing magnitude. Symptoms of these lethal coronaviruses differ.

MERS is a disease predominantly affecting the lower respiratory tract, which in most patients leads to pneumonia. Clinical manifestations are fever, malaise, chills, myalgia, cough, dyspnea, diarrhea, vomiting, and abdominal pain. In severely ill patients dyspnea is severe with acute respiratory failure, renal failure, and shock. As in SARS-CoV-2, there is a high incidence in older patients. Predictors of poor outcome include age above 60 years, male gender, diabetes mellitus, chronic lung disease and chronic renal disease, low albumin level and progressive lymphocytopenia [8]. MERS-CoV infections can be asymptomatic in 12.5–25% of patients [8].

SARS can present with hypoxia, cyanosis, fever, dyspnea and acute respiratory failure. The WHO case definition (2003) includes the following: (1) fever higher than 38 °C or history of such in the past 2 days, (2) radiological evidence of new infiltrates consistent with pneumonia, (3) chills, cough, malaise, myalgia, or known history of exposure, and (4) positive test for SARS-CoV by one or more assays.

In SARS patients, neutralizing antibodies are detected 2–3 weeks after the onset of disease, and 90% of patients recover without hospitalization [2]. About 10% of SARS patients develop severe respiratory failure after 5–7 days following infection, with interstitial pneumonia characterized by progressive diffuse alveolar damage.

In COVID-19, most frequent co-morbidities are hypertension, cardiovascular disease, diabetes, and obesity [9]. Age appears to be the strongest predictor of COVID-19 related death. Clinical manifestations of COVID-19 include fever, malaise, myalgia, non-productive cough, dyspnea, nausea, vomiting and diarrhea. Gastrointestinal symptoms can be the first manifestation of COVID-19, especially in patients with immunosuppressive drugs. Olfactory and/or gustatory dysfunctions have been reported in 64% to 80% of patients [10].

COVID-19 can progress to severe organ dysfunction of the heart, brain, lung, liver, kidney, and coagulation system [10], and can lead to myocarditis, cardiomyopathy, ventricular arrhythmias, and hemodynamic instability [10]. In severe infection, patients may develop acute cerebrovascular disease and encephalitis [10]. Hypercoagulopathy leading to both venous and arterial thromboembolic events occur in 10% to 25% in hospitalized patients, and in ICU patients with COVID-19 in 31–59% [10]. Approximately 72% of non-surviving COVID-19 patients had hypercoagulopathy [9].

SARS-CoV-2 also can induce vascular damage, and pre-existing endothelial dysfunction combined with the direct assault of SARS-CoV-2 on the vascular system may account for a high mortality of COVID-19 patients [9].

Hospitalized patients with COVID-19 need ICU treatment in approximately 17–35% of patients, most commonly due to hypoxemic respiratory failure requiring intubation and mechanical ventilation [10].

About 4–32% of patients are completely asymptomatic. However, it is unclear which of the following three scenarios are represented in these reports: (1) truly asymptomatic infection by individuals who never develop symptoms, (2) transmission by individuals with very mild symptoms, or (3) transmission by individuals who are asymptomatic at the time of transmission but subsequently develop symptoms [10].

### 1.3. Treatment of Coronaviruses

There is an urgent need to develop therapeutic drugs and vaccines against SARS-CoV-2 for the treatment and prevention of COVID-19, respectively. As these drugs still are under investigation, research also focuses on existing drugs with proven effectiveness in other (corona-)viral diseases. Sometimes this is referred to as "repurposing". Using currently approved drugs for other indications reduces time, costs and safety issues. Calcineurin inhibitors (CNI) showing favorable effects in multiple coronaviruses, thereby replacing the "one-drug-for-one-bug" paradigm. They are well-known, already existing drugs in transplant medicine used for solid organ transplant (SOT) recipients, and are also prescribed in rheumatology, dermatology and ophthalmology.

### 1.4. Scope of This Review

This review highlights the current evidence of CNI as pan-coronaviral inhibitors, including the current understanding of the biophysiological characteristics of CNI influencing the viral behavior in the human host. We also provide an outlook on what aspects should be considered when investigating this transplant medicine approach for the treatment of immunocompetent patients suffering from COVID-19.

## 2. Methods

We searched original papers, observational studies, case reports, and meta-analyses published between 2000 and 2020 in English in the PubMed database and Google Scholar using the keywords: (coronavirus), (treatment), (MERS), (SARS), (COVID-19), (tacrolimus), (ciclosporin), (cyclosporin) AND (calcineurin inhibitor). In addition to the commonly used preprint servers for COVID-19 research (bioRxiv and medRxiv, arXiv, Research Square, www.preprints.org, Open Science Framework, and the WHO Bulletin) we searched all available preprint servers mentioned in the preprint server directory ASAPbio registry (49 entries, last updated in January 2020) without indexing in Google Scholar for additional papers with the same search criteria as mentioned above. Namely, the following preprint servers were searched: Autorea, Cell Sneak Peek, Journal of Medical Internet Research preprints, Neuroimage Clinical First Look, Preprints with The Lancet, Social Science Research Network, Surgery Open Science first look, Therapoid, ViXra. We searched for additional references in the bibiographies of the detected papers to obtain additional references relating to the main topic. We also tried to obtain pre-clinical and clinical safety data from the two pharmaceutical companies that currently market the two main approved CNI drugs without responses.

We excluded studies in patients with clear indications for immunosuppressive therapy, such as solid organ transplant recipients or rheumatological patients, since we aimed to investigate the CNI in immunocompetent patients.

## 3. Results

Ten studies were included. Calcineurin inhibitor therapy has been documented to be effective in various coronaviral diseases both in vitro as well as in vivo. So far, no data in immunocompetent patients on effects of CNI in human SARS-CoV-2 infections have been published. The results are shown in Table 1.

**Table 1.** Coronaviral serotypes and treatment with calcineurin inhibitors.

| Coronaviral Serotype Studies in Humans | CNI | Remarks | Ref. No. |
|---|---|---|---|
| MERS-CoV | Tac | renal transplant recipient on tacrolimus survived | [11] |
| MERS-CoV | CsA | inhibition of viral replication | [12] |
| **Coronaviral Serotype Studies in Animals** | **CNI** | **Remarks** | **Ref. No.** |
| feline CoV | CsA | inhibition of viral replication in dose-dependent manner | [13] |
| turkey CoV | CsA | enhanced virus titers in kidney | [14] |
| **Coronaviral Serotype Studies In Vitro** | **CNI** | **Remarks** | **Ref. No.** |
| MERS-CoV | CsA + IFN-$\alpha$ | inhibition of viral replication | [15] |
| MERS-CoV, SARS-CoV | ALV | inhibition of viral replication | [16] |
| SARS-CoV, CoV-229E | CsA | inhibition of viral replication SARS-CoV replication impaired, but not fully blocked (1–5% of cells remained SARS-CoV positive, even in high CsA concentrations) | [17] |
| CoV-NL63, CoV-229E, SARS-CoV | CsA | inhibition of viral replication | [18] |
| SARS-CoV, CoV-NL63, CoV-229E | Tac | inhibition of viral replication | [19] |
| CoV-NL63 | CsA-d | inhibition of viral replication by CsA derivatives (Alisporivir, NIM811) | [19] |
| SARS-CoV-2 | CsA | potent antiviral activity in SARS-CoV-2, cyclophillin depedent (and calcineurin independent) | [20] |

ALV = alisporivir, CNI = calcineurin inhibitor, CoV = coronavirus, CsA = cyclosporin A, CsAd = cyclosporine A derivatives, IFN-$\alpha$ = interferon alpha, MERS = Middle East respiratory syndrome, Tac = tacrolimus.

The available studies with in vivo data on SARS-CoV, MERS-CoV and SARS-CoV2 include studies performed on animal models of coronavirus-related diseases. SARS-CoV replication has been studied in mice, Syrian golden and Chinese hamsters, civet cats, and non-human primates [21], and MERS-CoV in mice, camelidae and non-human primates [21]. These animal studies investigated protease inhibitors, monoclonal and polyclonal antibodies, convalescent plasma, interferons, ribavirin, lopinavir/ritonavir in both SARS-CoV and MERS-CoV [21], but to our knowledge there are no animal data on SARS-CoV and SARS-CoV-2 addressing CNI therapy. Exceptions are the feline and turkey CoV, which do not have the possibility of spillover into human hosts [13,14]. The other exception is alisporivir, a nonimmunosuppressive cyclophillin inhibitor (CsA analog) in SARS-CoV, with strong in vitro dose-dependent antiviral properties against SARS-CoV-2 [16].

## 4. Discussion

So far, the COVID-19 pandemic has led to more than 17 million SARS-CoV-2 infected patients and over 700000 deaths [7]. The number of cases is still, or again, on the rise in many countries and there currently is no geographic region where the pandemic seems totally under control.

### 4.1. COVID-19 in Solid Organ Transplant Recipients

Surprisingly, in solid organ transplant (SOT) recipients, a relatively low number of patients have been reported, in case reports or small case series [22]. Symptoms of COVID-19 in SOT recipients often can be atypical, such as gastrointestinal (i.e., diarrhea, anorexia, and upper abdominal discomfort) or neurological (i.e., delirium), and therefore this diagnosis needs a high index of suspicion [23].

Whilst SOT recipients require life-long administration of immunosuppressive drugs in order to minimize alloreactivity and preserve solid organ allograft function, severe infections related to immunosuppression are feared. Based on this, SOT recipients have been considered to belong to the vulnerable population for SARS-CoV-2 infections and severe consequences of COVID-19 were expected.

Paradoxically, SOT recipients with SARS-CoV-2 infections have shown a relatively benign course of disease, most of them with a favorable outcome within a short timeframe. D'Antiga (Italy) published on March 20, 2020 the first descriptive analysis of clinical observations in SARS-CoV-19 positive transplant patients and suggested that unlike common viral agents (e.g., adenovirus, influenza, respiratory syncytial virus), infection with SARS-CoV-19 might not lead to a worse general condition in immunosuppressed patients [24–26]. Another study showed that transplant status was not associated with COVID-19 mortality [27].

This is in sharp contrast to many immunocompetent COVID-19 patients, in whom a subset develop severe COVID-19 which is associated with a high mortality rate.

Moreover, the number of SOT recipients with COVID-19, described in several case reports and some case series, is relatively low compared to the number of immunocompetent COVID-19 patients. Although definite numbers of SOT recipients with COVID-19 have not been reported by the transplant societies, our own estimation is that there are less than 1000 patients. However, these numbers are still increasing as the pandemic is ongoing. Long-term consequences of COVID-19 in SOT recipients cannot be estimated yet due to the relatively short follow-up duration of a few months.

### 4.2. Cytokine Storm Syndrome (CSS)

In the severe COVID-19 phase, the pathophysiological response is a cytokine-mediated systemic hyperinflammation, called cytokine storm syndrome (CSS) or cytokine release syndrome (CRS). CSS is a life-threatening emergency associated with high mortality. It was first described in renal allograft recipients [28], receiving the anti-T-cell antibody muromonab-CD3 (OKT3), an immunosuppressive drug [28]. These patients developed systemic reactions that closely resembled the symptoms induced by the injection of pure recombinant cytokines [28], which was related to a massive release of highly biologically active mediators [28]. In CSS, laboratory results demonstrate pancytopenia (anemia, leukocytopenia, thrombocytopenia), coagulation disorders, elevated serum creatinine, liver enzymes, C-reactive protein (CRP) and hyperferritinemia.

The presence of hyperferritinemia seems to play a relevant pathophysiological role in CSS. In autoimmune diseases, such as rheumatoid arthritis, systemic lupus erythematosus (SLE) and the anti-phospholipid syndrome, it is a well-known feature [29]. H-ferritin has been shown to suppress myeloid cells, and also affects lymphoid cells by suppressing the proliferation of T-cells and impair the maturation of B-cells [30]. Moreover, ferritin may favor the loss of tolerance and the onset of autoimmunity [30]. Ferritin can be also a pro-inflammatory signaling molecule [29]. Hyperferritinemia has been associated with different CSS-related conditions such as macrophage activation syndrome (MAS) and septic shock [29].

CSS is clinically characterized by persistent fever, lymphadenopathy, hepatosplenomegaly, central nervous system (CNS) abnormalities with multiple organ failure (MOF), disseminated intravascular coagulation (DIC), renal and/or cardial insufficiency, and shock.

In COVID-19, the CSS spectrum of respiratory symptoms is wide, and can be mild (cough, mild dyspnea) to severe (severe dyspnea, with progression to Acute Respiratory Distress Syndrome (ARDS) or Acute Lung Injury (ALI), with a fulminant post-ARDS pulmonary fibrosis. The pulmonary complications in CSS has an aerogenic and a vascular route. The first is by aerogenic SARS-CoV-2 transmission which leads to SARS-CoV-2 reaching the ACE-2 receptors in the alveolar epithelial cells. This results in the downregulation of ACE-2 expression and increasing the angiotensin level, leading to increased pulmonary capillary permeability and pulmonary edema [31]. The second is by the blood circulation, where SARS-CoV-2 reaches the lung again, interacting with the ACE-2 receptors on the surface of alveolar capillary endothelial cells, where it attacks the capillary endothelium. The resultant

immune responses further aggravate lung injury by the CSS [31]. These cytokines include interleukin (IL)-1β, IL-1Rα, IL-2, IL-10, fibroblast growth factor (FGF), granulocyte-macrophage colony stimulating factor (GM-CSF), granulocyte-colony stimulating factor (G-CSF), interferon-γ-inducible protein (IP10), monocyte chemo attractant protein (MCP1), macrophage inflammatory protein 1 alpha (MIP1A), platelet derived growth factor (PDGF), tumor necrosis factor (TNFα) and vascular endothelial growth factor (VEGF) [32]. Moreover, in severe COVID-19, there is a reduction of natural killer (NK) cells, CD4+ and CD8+ T-lymphocytes and IFN-γ expression in CD4+ T-lymphocytes. In this phase, SARS-CoV-2 infection leads to a reduction and functional exhaustion of T cells [33] and by the above described mechanisms, SARS-CoV-2 is hijacking our immune defense systems.

These inflammatory factors may be among the leading causes in the rapid worsening of COVID-19. Similar to SARS-CoV and MERS-CoV, in SARS-CoV-2 increased amounts of proinflammatory cytokines in serum were associated with pulmonary inflammation and extensive lung damage [34]. Moreover, compared to patients with less severe disease, patients requiring ICU admission had higher serum cytokine concentrations than those that did not require ICU admission, suggesting that the cytokine storm was associated with disease severity [34]. However, regarding the role of cytokine storm in COVID-19, it still is not clear which cytokine(s) plays a critical role in the initiation of severe COVID-19 [35].

Another major unanswered question is why most COVID-19 patients with CSS are elderly patients, and CSS is extremely rare in young COVID-19 patients. One explanation might be that aging is associated with mild elevated levels of local and systemic pro-inflammatory cytokines, including IL-6, IL-8, TNF-α, IL-13, IFN-γ, as well as acute phase proteins. This chronic mild inflammation in aging, so-called "inflamm-aging" [36], results in an increased risk of a cytokine storm in some critical elderly patients with COVID-19 infection [36].

### 4.3. Cytokine Storm Syndrome in Other Diseases

The cytokine storm syndrome has been described in infectious and non-infectious diseases, and is not unique to COVID-19. Cytokine profiles can be slightly different, dependent on the cause of the CSS, as has been reviewed by Gao et al. [31]. It has also been observed in other viral infections (SARS-CoV, MERS-CoV, Avian H5N1 Influenza and the Gram-negative bacterium *Francisella tularensis*), graft-versus-host disease, autoimmune diseases (SLE, systemic juvenile idiopathic arthritis), hematologic conditions (hemophagocytic lymphohistiocytosis) and medications [37]. It is possible that increased levels of proinflammatory cytokines in older people are responsible for a more severe course of the disease or a particular aspect of immunosenescence [38].

Although the human immune response against SARS-CoV-2 remains poorly defined, it has been suggested that calcineurin inhibitors, used chronically in many solid organ recipients, may play a protective role in patients with COVID-19 [39–41].

### 4.4. The Calcineurin/NF-AT Signaling Pathway

To understand the mechanisms of CNIs, the interaction between intracellular calcineurin and nuclear factor of activated T cells (NF-AT) is important (Figure 1). In resting cells, NF-AT proteins are hyperphosphorylated and are localized in the cytoplasm. On activation, NF-AT proteins undergo rapid dephosphorylation by calcineurin. The dephosphorylated NF-AT proteins then translocate into the nucleus, where they regulate gene transcription. NF-ATs regulate a large number of inducible genes in immune cells, including cytokines, cell-surface receptors, and chemokines [42].

Dephosphorylating NF-AT by calcineurin is a calcium-dependent process, and as soon as the calcium signals cease, it leads to rephosphorylation, initiating NF-AT to return from its active state in the nucleus to its inactive state in the cytoplasm.

### 4.5. Two Calcineurin Inhibitors in Clinical Use

Currently, cyclosporin A (CsA) and tacrolimus (FK506) are CNIs used in the clinical setting. CsA was approved by the FDA in 1983. It is a cyclic polypeptide, derived from the fungus

*Tolypocladium inflatum* [43]. Tacrolimus (also known as FK506), used since 1995, is a macrolide antibiotic, isolated from the soil bacterium *Streptomyces tsukabensis*, with quite a similar mechanism of action as CsA [44], which will be discussed later.

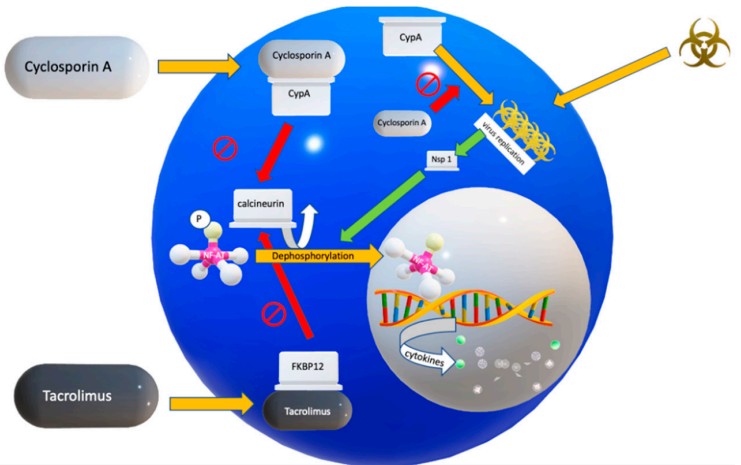

**Figure 1.** Pathogenesis of COVID-19 and the suggested actions of cyclosporin and tacrolimus. Upon binding of a CoV protein to cyclophilins, the Calcineurin/nuclear factor of activated T cells (NF-AT) pathway is activated, via the coronaviral non-structural protein-1 (Nsp) leading to a systemic cytokine storm. In addition, cyclophilin A (CypA) probably stimulates CoV replication. Cyclosporin A inhibits viral replication. Adapted from Tanaka et al., *Viruses* **2013**, *5*, 1250–1260; doi:10.3390/v5051250.

## 4.6. Mechanism of Action of Cyclosporin A (CsA)

As a calcineurin inhibitor, cyclosporin A (CsA) binds to cyclophilins (Cyps). Cyps are the binding partner proteins of CsA, which, as host cell receptors for CsA, mediate the immunosuppressive action of CsA, by inhibiting calcineurin (Figure 1) [11]. Cyclophilins belong to the immunophilins, and the most abundant cyclophilin is Cyclophilin A (CypA), which is widely distributed in almost all tissues and accounts for 0.1–0.4% of the total protein content in a cell [45]. It also is abundant in the cytosolic extract from lymphocytes, and has a high affinity for CsA. CypA acts as an acceleration factor in protein folding and assembly [45].

Inflammatory stimuli [46,47], oxidative stress [46,48] and activated platelets [49] result in cellular secretion of CypA. Following stimulation with reactive oxygen species (ROS), CypA was detectable at the plasma membrane of vascular smooth muscle cells within 30 min of stimulation [46]. When CsA inhibits Cyps, several important effects occur, which are summarized below:

(a) Preventing the Cytokine Storm Syndrome

First of all, as described before, CNIs block the translocation of NF-AT from the cytosol into the nucleus. Inhibiting the NF-AT dephosphorylation inhibits expression of NF-AT dependent genes. This is thought to be the major mechanism in preventing the systemic cytokine storm syndrome in COVID-19. Blockade of NF-AT into the nucleus prevents transcription of genes that encode for cytokines such as interleukins (IL) [11], and in this way inhibits the pro-inflammatory pathway.

(b) Direct Antiviral Effect of CsA

CsA may have a direct antiviral effect, in which Cyps might play a critical role in replication of coronaviruses [11]. Although in vitro data do not necessarily imply effects in vivo, there are numerous in vitro data suggesting antiviral properties of CsA in other non-coronaviral diseases. Through blocking the interaction of cellular cyclophilins with viral proteins and inhibiting viral RNA synthesis, it inhibits replication in hepatitis B virus, hepatitis C virus (HCV), HIV virus, influenza A virus, West Nile virus, Rift Valley fever virus, and Zika virus [50]. In HCV, clinical trials have shown that even non-immunosuppressive derivatives of CsA still can potently suppress HCV viral load in patients [51,52].

(c) Indirect Effect of CsA in Cardiovascular Complications of COVID?

CypA might be interesting for the understanding and possibly treatment of cardiovascular morbidity in COVID-19. CypA is also a growth factor for vascular smooth muscle cell (VSMC) under oxidative stress [45] and in this way plays a crucial role in cardiovascular disease. Whether this mechanism is relevant in COVID-19, and whether CsA could play a protective role in this context, is currently unknown. Nevertheless, it would be interesting to investigate CypA as a biomarker in cardiovascular complications of COVID-19. Alternatively, it might explain the elevated risk of COVID-19 patients for cardiovascular comorbidity, since these patients have elevated reactive oxygen species (ROS), which induce secretion of CypA from VSMC.

(d) Direct Antifibrotic Effect of CsA

Pulmonary fibrosis is one of the major complications in COVID-19 patients [38], with acute respiratory distress syndrome (ARDS) being the main cause of post-COVID pulmonary fibrosis. Similar cytokine profiles in idiopathic pulmonary fibrosis (IPF) and COVID-19 suggest analogous pathomechanisms in both diseases. Interestingly, cytokine overexpression in IPF, COVID-19 and SARS/MERS all show elevated IL-1B, IL-6, MCP1, TNF-$\alpha$ and TGF-$\beta$ [38]. Therefore, drugs useful in the treatment of IPF could also be beneficial for COVID-19 patients [38]. CsA might have a direct antifibrotic action, as has been demonstrated in patients with antisynthetase syndrome associated interstitial lung disease, who were refractory to corticosteroids, but improved on CsA [53,54]. In tacrolimus, the combination with methylprednisolone pulse therapy showed to mitigate acute exacerbations (AE) of IPF, prevented re-AE IPF, and contributed to a better prognosis compared to steroid monotherapy [55]. Studies have shown that CsA is superior to the corticosteroid monotherapy in terms of prognosis for IPF [56,57].

This feature of CsA may be an important aspect to consider when attempting to prevent post-COVID-19 pulmonary fibrosis, although further studies are still needed to elucidate the magnitude of the effect.

*4.7. Mechanism of Action of Tacrolimus*

Tacrolimus (FK506) is another important calcineurin inhibitor, known from transplant medicine. In contrast to CsA, which binds to the immunophilin Cyp, tacrolimus binds to the immunophilin called FK-506 binding protein (FKBP12) (Figure 1). Due to tacrolimus, the FKBP forms a complex with the calcium-dependent phosphatase named calcineurin and inhibits the activity of calcineurin. Similar to CsA, tacrolimus has also been shown to have a favorable activity as antifibrotic agent, for example in patients with idiopathic inflammatory myopathy or antisynthetase syndrome- interstitial lung disease (ILD) [53,58]. Interestingly, in animal experiments, upregulated FK506-binding protein 10 (FKBP10) in bleomycin-induced lung fibrosis and IPF improved with knockdown of FKBP10, attenuating collagen secretion [59]. Although no firm conclusions can be drawn yet, tacrolimus also might have an anti-fibrotic activity in COVID-19.

Although clinical trials are still awaited, preliminary clinical experience reports suggest that tacrolimus is protective for liver transplant recipients, but so far not for other organs, for example kidneys. Concerning this observation in tacrolimus, an intriguing question is, if tacrolimus has a different degree of protection against CSS effects in various organs. In other words, is there possibly a specific organ effect, beyond the pharmacokinetics? The various cytokines may play different roles in solid organs. The most important cytokines resulting in CSS in the liver, may be different from the cytokines that are the most important ones in the kidney or lung. This is however still speculative. Nevertheless, in other (non-coronaviral) diseases, the different CNIs (CsA and tacrolimus), when compared to each other, were shown to be discordant in respect to the antiviral effects. This could be the result of the different cytokine profiles addressed by the different CNIs, leading to protection of different organs. Moreover, ACE2 is abundantly present in the lung epithelial cells, and CNIs only have antiviral effects after infection of the target cells, when the virus replication starts in the

cytoplasm. Different ACE2 density could contribute to differences in organ protection by CNIs, called the immunolocalization of ACE2. Another study suggested a different receptor repertoire potentially involved in the SARS-CoV-2 infection at the epithelial barriers and in the immune cells [60]. As mentioned before, the understanding of the pathogenesis is still incomplete.

*4.8. Alternative Drugs to Inhibit the Cytokine Storm Syndrome*

Among alternative drugs that may have the ability to inhibit the systemic hyperinflammation in COVID-19 are IL-6 blockers (Tocilizumab, Sarilumab), IL-1 blocker (Anakinra, Canakinumab), heparins (low molecular weight and unfractionated heparin), intravenous immunoglobulins (IVIG), hyperimmune immunoglobulins (neutralizing antibodies), JAK inhibitors (Ruloxitinib, Bariticinib), corticosteroids (methylprednisolone, dexamethasone), statins and recombinant human angiotensin-converting enzyme 2 (rh ACE2). These are discussed elsewhere [61].

Interestingly, there is another CypA inhibiting drug, without immunosuppressive activity, named alisporivir [16,62,63]. It is a non-immunosuppressive analogue of CsA with strong Cyps inhibition properties. Alisporivir has reached phase three clinical development for the treatment of COVID-19 [63].

Preclinical data show strong antiviral and cytoprotective properties of alisporivir in various models of coronavirus infection, including SARS-CoV-1, MERS-CoV and SARS-CoV-2 [63]. Nevertheless, an important question remains if alisporivir can also inhibit the cytokine storm syndrome, as it has no immunosuppression activity.

## 5. Conclusions

Calcineurin inhibitors have been proven to be effective in a number of coronaviral diseases and other related conditions. The most important CNIs used in the clinical context are cyclosporin and tacrolimus. They block the calcineurin pathway by forming complexes with immunophilins, being cyclophilin for cyclosporine A, and FKB12 for tacrolimus. These immunophilins prevent calcineurin from dephosphorylating the NF-AT transcription factor. This results in the inhibition of the transcription of genes encoding for cytokines, decreasing the risk of CSS.

Paradoxically, the CNIs that are crucial to solid organ transplantation and render SOT recipients more susceptible to opportunistic infections, appear to also have the ability to suppress the cytokine storm syndrome in COVID-19. In this regard it would not be rational to follow the guidelines of the Massachusetts General Hospital (Boston, MA, USA), which advises to consider decreasing tacrolimus/cyclosporin by 50% in solid organ transplant recipients with COVID-19, and in critical illness in liver and kidney transplant recipients to stop all immunosuppression except for prednisolone.

In the search of effective treatment options for the novel coronavirus SARS-CoV-2, CNIs should be evaluated as a first line treatment option because of the suggested direct antiviral effects as and its potential to suppress the severe systemic hyperinflammation state and thus reduce the disease severity of COVID-19 [64]. Based on the known CNI effects in various coronaviral diseases, they are likely to be effective in multiple coronaviral serotypes (including SARS-Cov-2) and, as multitarget agents, may more effectively reduce the likelihood of developing viral resistance as compared to other strategies. If CNIs can be proven to be effective also for previously immunocompetent patients with moderate to severe COVID-19, then they may be an easy and affordable option for the rapid management of the COVID-19 patients in many parts of the world, since these drugs are affordable and already quite widely available.

## 6. Outlook

Given the unique mechanism in mitigating the cytokine storm syndrome, the calcineurin/NF-AT signaling pathway presents an attractive target for therapeutic drug development for prevention of severe COVID-19. Currently the research of CNI for SARS-CoV-2 infection and prevention of severe COVID-19 disease is still limited. Therefore, the results of the clinical TACROVID trial from

Barcelona, Spain are urgently awaited and currently pending. This trial investigates the clinically important question of tacrolimus in patients with COVID-19, with in one arm treating patients with methylprednisolone pulses 120 mg/day for three consecutive days (if they were not previously administered) with tacrolimus at the necessary dose to achieve plasma levels of 8–10 ng/mL, versus the other arm with usual care including all necessary treatments with the exception of CNIs [65].

Recruitment has also started for the study in which cyclosporine is clinically tested in patients with COVID-19 requiring oxygen supplementation but not requiring ventilator support [60]. This trial is a phase 1 safety study to determine the tolerability, clinical effects, and changes in laboratory parameters of short course oral or IV cyclosporine administration [66].

Major questions that remain open should be addressed in research, and the TACROVID trial, the American cyclosporine study (and probably an alispirovir trial in the near feature) will likely shed more light on this issue [65,66]. In our opinion, clinically relevant questions comprise those mentioned in Table 2.

**Table 2.** Proposed research questions on treatment with calcineurin inhibitors.

| Research Question | Possible Answers in Literature | Refs. |
|---|---|---|
| Which patients with COVID-19 could benefit from the addition of CNI to the standard therapy | • The inclusion criteria of the TACROVID trial could be helpful. They include COVID-19 infection confirmed by PCR, new onset radiological infiltrates, respiratory failure (PaO2/FiO2 < 300 or satO2/FiO2 < 220), C-reactive Protein > 100 mg/L and/or D-Dimer > 1000 μg/L and/ or Ferritin > 1000 ug/L. | [65] |
| Does CypA play a role in cardiovascular morbidity in COVID-19 patients? | • Could CypA be a marker for cardiovascular morbidity in COVID-19 patients? | [45] |
| How to screen for patients with a high risk of progression to more severe stages of COVID-19 and thus merit pharmacological interventions | • Several scoring systems are available, such as the AIFELL score, which includes an altered sense of smell/taste, inflammation (C-reactive protein ≥ 30 mg/L), radiological infiltrates, fever (≥38.0 °C), elevated lactate dehydrogenase (LDH) levels (>400 U/L) and lymphocytopenia (absolute count < 1.45 G/L). The score is calculated by counting the number of criteria met at initial presentation in the emergency room, whereas each criterion equals one point (Score range 0 to 6 points). | [67] |
| Which patients with COVID-19 should be excluded from CNIs? | • life expectancy ≤ 24 h, glomerular filtration ≤ 30 mL/min/1.73 m$^2$, leukopenia ≤ 4000 cells/μL. (exclusion criteria in TACROVID trial) | [65] |
| CNI monotherapy or combination therapy with either a corticosteroid, an antimetabolite (Mycophenolate) | • dexamethasone led to a lower 28-day mortality among those who were receiving either invasive mechanical ventilation or oxygen alone at randomization but not among those receiving no respiratory support. Would it also improve the effect of CNIs? | [68] |
| Alternative immunomodulatory drugs? | • Rapamycin (m-TOR inhibitor)? Probably yes<br>• Rapamycin (m-TOR inhibitor)? No<br>• Many other immunomodulatory drugs are reviewed elsewhere | [69]<br>[70]<br>[71] |
| Alisporivir as non-immunosuppressive cyclophilin inhibitor? | • Inhibition (in vitro) of SARS-CoV-2 in literature<br>• However, when not immunosuppressive, does it protect against cytokine storm? Or only protection against the cytopathic effect? | [62,63] |

From a general point of view there is a vast experience with CNI in transplantation medicine including dosing regimens and experience with achieving specific drug levels by therapeutic drug monitoring. This experience can be beneficial when, within a short time frame, a CNI-based immunosuppression should be established, which also takes into account comorbidity (renal function, other medication/interactions). Which target drug levels should be used is another open question. Based on the experience with SOT recipients, similar drug levels should probably be targeted as for maintenance of immunosuppression in such patients. Whether in certain situations augmentation of immunosuppression may be wise as a second step would have to be evaluated as well. In addition to the immunosuppressive strategy the most appropriate marker for disease activity measurement in these COVID-19 patients will have to be determined in the context of the immunosuppressive therapy (CRP, Procalcitonin (PCT), certain Interleukins, differential white blood cell count, etc). Besides dose and monitoring of the immunosuppression, the duration of continuation of this therapy will have to be evaluated in addition to the effects on viral load and potentially observed prolonged viral shedding. The best evidence is probably derived from dual or triple immunosuppressive regimens in SOT, therefore the combination of two immunosuppressive drugs which certainly includes a CNI because of its pleiotropic effects (including antiviral effects) is likely to be a promising pharmacological strategy to prevent severe COVID-19. As a potential predictor of severe disease course, the AIFELL

score may be considered [67]. The triage score relies on disease markers that at an early stage indicate whether a more severe disease progression may be expected.

Although there remain many open questions, CNI should be investigated as a first treatment option, based on evidence of direct antiviral effects and its properties preventing CSS, as has been observed in COVID-19 with predominantly immunopathological changes of the respiratory tract.

**Author Contributions:** Conceptualization, R.H.; M.M.S.; writing—original draft preparation, R.H.; writing—review and editing, R.H.; C.S.; F.G.; M.M.S.; supervision M.M.S. All authors have read and agreed to the published version of the manuscript.

**Funding:** This research received no external funding.

**Conflicts of Interest:** The authors declare no conflict of interest.

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
