# Peer review of "Transplant Drugs against SARS, MERS and COVID-19"

_2673-3943, doi:10.3390/transplantology1020007_

Round 1
Reviewer 1 Report
I congratulate the authors for this relevant study on the mechanisms by which CNIs might prevent coronavirus disease in humans. Clinical trials are awaited, but preliminary clinical experience reports that tacrolimus is protective for liver transplant recipients, but not for other organs, as for example kidneys. What do the authors think as a possible cause? Could they explain more in details if beyond the pharmacokinetics, there is also an organ effect?
Author Response
Thank you for this comment.
We have added a possible explanation.
Although clinical trials are still awaited, preliminary clinical experience reports suggest that tacrolimus is protective for liver transplant recipients, but so far not for other organs such as for example kidneys. Concerning this observation in tacrolimus, an intriguing question is, if tacrolimus has a different degree of protection against CSS effects in various organs. In other words, is there possibly a specific organ effect, beyond the pharmacokinetics? The various cytokines may play different roles in solid organs. The most important cytokines resulting in CSS in the liver, may be different from the cytokines that are the most important ones in the kidney or lung. This is however still speculative. Nevertheless, in other (non-coronaviral) disease, the different CNIs (CsA and tacrolimus), when compared to each other, were shown to be discordant in respect to the antiviral effects. This could be the result of the different cytokine profiles addressed by the different CNIs, leading to protection of different organs. Moreover, ACE2 is abundantly present in the lung epithelial cells, and CNIs only have antiviral effects after infection of the target cells, when the virus replication starts in the cytoplasm. Different ACE2 density could contribute to differences in organ protection by CNIs, called the immunolocalization of ACE2. Another study suggested a different receptor repertoire potentially involved in the SARS-CoV-2 infection at the epithelial barriers and in the immune cells[54].
Reviewer 2 Report
The paper is an interesting review on the topic of the use of immunosuppressive drugs in the treatment of severe Coronavirus diseases.
The topic is generally well addressed, but it is not clear in the Results section which papers were evaluated.
In particular shoudl perhaps clarify the sentence “We excluded studies in patients with clear indication for immunosuppressive therapy”.
The Authors included some papers relative to the outcome of COVID 19 infected transplanted patients and it is not clear how they chose those papers and not others on the same issue. Moreover some papers evaluating the outcome of different immunosuppressed patients like rheumatologic or dermatologic treated with cyclosporine could be analyzed.
The acronym IPF should be introduced earlier in the text
Author Response
Response a)
Thank you for this comment. We have amended the search strategy to clarify all sources accessed and we have also changed the sentence by adding the highlighted text: We excluded studies describing patients with a clear indication for immunosuppressive therapy, such as solid organ transplant recipients, because our main focus was CNI use in immunocompetent patients and antiviral effects of these compounds.
Response b)
This is a good suggestion. As mentioned in response a) we amended the text to define better what sources we searched for papers or preprints. We included our paper which was published in Transplantology because it describes examples of COVID-19 infections in different organ transplant recipients and illustrates these cases for the reader. Patients with immunosuppression like rheumatologic or dermatologic patients have been described in other papers, and are indeed interesting. However, in order to serve the readers of "Transplantology", articles on transplanted patients are more suitable for our review. Moreover, it is not our scope to elaborate on immunosuppressed patients, since we would like to answer the question whether CNIs could be suitable for immunocompetent patients with COVID-19. Treating immunocompetent patients with immunosuppression would be a totally new paradigm.
Response c)
We introduced the acronym IPF earlier in the text “Pulmonary fibrosis is one of the major complications in COVID-19 patients[31], with acute respiratory distress syndrome (ARDS) being the main cause of post-COVID pulmonary fibrosis. Similar cytokine profiles in idiopathic pulmonary fibrosis (IPF)) ...etc.”
Reviewer 3 Report
The authors present a very attractive hypothesis that calcineurin inhibitors, used in transplant recipients, are potential treatment for COVID-19. The Introduction, and Discussion are well-written, and clear. The Material and Methods, as well as Results, are the main disadvantages of the paper. They are perfunctory and contain little content. Additionally, Discussion is only remotely linked with the Results. Arguments for utility of calcineurin inhibitors in therapy of COVID-19 are quite poorly associated to the reported results. I rather suggest to publish the work as a description of a medical hypothesis.
Author Response
Response a+b) This is a valid comment. We have extended the information concerning our search strategy and provided more details what sources searched in order to obtain information on the topic of interest. We also searched preprint servers which were not indext in Google Scholar (see methods section). Since at the time of our investigation there was little information on SARS-CoV-2 experimental and clinical data, we resorted to searching for data relating to similar coronal virus infections with the intention to extrapolate from the available data to COVID-19. Where recent data was available to SARS-CoV-2, we naturally included this information, even if it originated from in vitro, in vivo (animal) or just case reports. We mentioned the TACROVID study already in our review and in the meantime a second study (with ciclosporin) is underway investigating the hypothesis outlined in this review. The reference of the CsA study has now been added.
Response c) Your comment is well taken and certainly merits consideration.
We emphasized the hypothetical character of our manuscript with the sentence "The aim of this narrative review is to present the current evidence of CNI in coronaviral diseases, the biophysiology of CNI and to suggest possible ways to study CNI as new treatment for COVID-19". In the meantime, these studies have started (the Spanish TACROVID study with tacrolimus, and the American study with cyclosporine). This means that the CNI hypothesis is being tested right now in the clinical setting with immunocompetent COVID-19 patients. At the moment of submitting our manuscript, the American trial had not started yet, but since 1st of September 2020 it has been recruiting patients. We therefore included this studyl in our references and the text of our manuscript: "Also recruitment has started for the study in which cyclosporine is clinically tested in patients with COVID-19 requiring oxygen supplementation but not requiring ventilator support[60]. This trial is a phase 1 safety study to determine the tolerability, clinical effects, and changes in laboratory parameters of short course oral or IV cyclosporine administration[60]."
Round 2
Reviewer 3 Report
The article was improved.